# Experts' views on how to design a tobacco control fund in the UK

Shona Hilton  , Marissa J Smith , Christina H Buckton, Chris Patterson

## ABSTRACT

**Objective** To explore expert views on the potential value, and approaches to establishing and administering a tobacco control fund in the UK.

**Design** Semistructured interviews and follow-up discussion groups.

**Subjects** Twenty-four UK and international experts on tobacco control regulation, public health, economics or law from the academic, public, private and third sector.

**Methods** Participants considered the relative merit of (1) general excise tax on retail tobacco sales; (2) ring-fenced hypothecation of excise taxes on retail tobacco sales; and (3) a direct levy on tobacco manufacturers. Preliminary synthesis of interview findings was deliberated on in two follow-up discussion groups to identify key considerations for policy design.

**Result** Most experts agreed that a ring-fenced tobacco control fund would be a valuable method of raising predictable and reliable funds from tobacco producers either using either companies' sales volume or market share as a way to establish the proportion they should pay. Experts predominantly recommended that a fund in the UK should be administered by a government body with devolved nation input and with an independent advisory group. They typically indicated that funding should be allocated yearly with a distribution at local, regional and national levels to support smoking prevention and cessation rather than treatment activities with priority given to measures that tackle smoking-related inequalities.

**Conclusion** There was overwhelming agreement by experts on the need to establish a tobacco control fund to help meet the proposed government tobacco-free targets to reduce adult smoking prevalence to 5% by 2030 (England) and 2034 (Scotland).

## STRENGTHS AND LIMITATIONS OF THIS STUDY

⇒ Methodology includes semistructured interviews with 24 UK and international experts on tobacco control regulation, public health, economics or law from the academic, public, private and third sector, facilitating understandings of the potential value of a tobacco control fund in the UK.

⇒ Follow-up discussion groups created informed dialogue between experts to collaboratively identify key considerations for policy design in this area by bringing together groups of policy actors diverse in terms of their specific areas of expertise and the sectors within which they have professional experience.

⇒ Quantitative thematic analysis of the data allows depth of opinions but cannot offer predictions about the frequency of specific opinions with a wider population.

⇒ The policy research offered new insights into an under-research area, but the complexity of policies and policy-making environments is such that transferring learning from one policy to a different policy is challenging.

MRC/CSO Social and Public Health Sciences Unit, University of Glasgow, Glasgow, UK

**Correspondence to**
Marissa J Smith;
marissa.smith@glasgow.ac.uk

## INTRODUCTION

Worldwide tobacco kills more than 8 million people per annum[1] including nearly 100 000 preventable deaths in the United Kingdom (UK).[2] Tobacco is highly addictive, there is no safe level of exposure and all forms of tobacco are harmful to health increasing the risk of cancers, heart disease and other Noncommunicable diseases (NCDs).[1] Despite a broad range of effective tobacco control policies, the tobacco trade continues to be highly profitable.[3] In contrast, the economic costs of tobacco use in society are greater than the costs for treating tobacco-related diseases.

For example, in the UK, revenue from excise duty on tobacco sales continues to be substantially lower than the health costs of smoking.[4] While UK smoking prevalence has declined precipitously in response to tobacco control action,[5 6] the smoking inequality gap has grown[6] as smoking contributes to poverty by diverting household spending from basic needs such as food and shelter to tobacco.

To further reduce smoking, the Westminster and Scottish governments have proposed targets to reduce adult smoking prevalence to 5% by 2030 and 2034, respectively.[7 8] However, for these proposed targets to be met prevalence rates need to decline at a much faster rate,[9] which may require additional tobacco control measures.

One policy option that has been proposed is to establish a ring-fenced tobacco control fund. This system for health promotion has been pursued in other countries including Australia, Vietnam, Korea and Thailand.[10 11] In the UK in 2015, Her Majesty's (HM) Treasury published their conclusions on an earlier

consultation on the potential design of a levy on tobacco manufacturers and importers.[12] The consultation considered a tobacco levy under the administration of the existing corporation tax system, imposed on manufacturers and importers of products on which tobacco excise duty is paid.[13] While the proposal received the support of a broad range of health charities, professional bodies and academics, the UK Government decided not to pursue the tobacco levy, citing concerns that costs would be passed on to consumers and that tobacco sales are already subject to escalating duties.[13] Since the government rejected this tobacco industry levy, other fiscal approaches to tackling the harms caused by unhealthy products have been introduced, including Scotland's minimum unit pricing for alcohol,[14] and the 'soft hypothecation' of the UK's soft drinks industry levy (SDIL).[15] In light of an increasing political willingness to implement other fiscal interventions, and the continued advocacy for a tobacco control fund from the public health community,[16 17] raising revenue for a tobacco control fund from the tobacco industry remains a viable policy option. This paper explores contemporary views of UK and international tobacco control and public health experts on the potential value of and approaches to establishing and administering a tobacco control fund. In doing so, we identify key considerations for its design.

## METHODS
### Interviews
We developed a purposive sampling frame to target UK and international experts in tobacco control regulation, public health, economics or law from the academic, public, private and third sector. Twenty-four experts agreed to participate after reading the participant information sheet, privacy notice and signing the consent form. Eighteen were based in the UK, four in the USA and two in South Africa. Table 1 illustrates the distribution of participants by the sector in which they primarily worked and their principal topic of work.

A semistructured interview schedule (online supplemental appendix A) was informed by reviewing international academic and grey literature on tobacco control funds. The interviews were conducted between September 2020 and January 2021 by CP and CHB. Only one interview was conducted by telephone and the remaining 23

interviews were conducted using Microsoft Teams video meetings. The interviews lasted between approximately 45 and 60 min, all were recorded and transcribed verbatim.

### Discussion groups
In March 2021, two follow-up online discussion groups were conducted by CHB and CP with nine individuals using Microsoft Teams. Participants were selected for these follow-ups based on their sectorial expertise and to represent key disciplines. The first discussion group (n=5) included three third sector professionals with expertise spanning tobacco control and public health advocacy and two academic economists. The second discussion group (n=4) included two public sector professionals with roles in tobacco control and public health policy and two academics with expertise in law and public health. The aim of these groups was to consider the synthesis of views from the interviews on the potential value of a tobacco control fund and to identify key considerations for policy design. Each discussion group lasted two hours, and group discussions were recorded for later checking against the minutes.

### Analysis
We conducted thematic analysis of the data form the interview transcripts and discussion group minutes. The process followed Braun and Clarke's[18] six-phase framework for thematic analysis. The research team read and re-read the transcripts to become familiar with the data, and then iteratively constructed a coding frame to enable consistent organisation of relevant data. NVivo was used to organise categories on the basis of inductive themes that emerged from close reading of the, capture of both areas of agreement and less typical perspectives across a range of categories. The discussion group recordings and minutes were cross-compared with the interview coding frame to confirm and expand on codes relating to recommendations for policy design of a tobacco control fund. Where appropriate, the number of participants that gave specific opinions are presented as counts and proportions to help illustrate the balance of opinion with the sample. However, it must be noted that, given the qualitative methodology used in this study, these numbers cannot necessarily be generalised to any wider population.

### Patient and public involvement
None.

## RESULTS
The results are presented in accordance with the inductive coding categories developed during the analysis stage.

### What is the potential value of a tobacco control fund?
There was general agreement that a tobacco control fund could be a valuable revenue for raising predictable and reliable funds direct from the tobacco industry. Typically, this was viewed as a way to boost current public health efforts:

**Table 1** Sample composition by primary sector of work and primary area of expertise

| Primary sector | Professional disciplinary approach to tobacco control | | | |
| | Economics/law | Public health | Other | Total |
|---|---|---|---|---|
| Academia | 6 | 3 | 0 | 9 |
| Public sector | 0 | 4 | 3 | 7 |
| Third sector | 1 | 1 | 5 | 7 |
| Private sector | 1 | 0 | 0 | 1 |
| Total | 8 | 8 | 8 | 24 |

The more money that we can earmark, ring-fence into public health and tobacco control efforts the better from a public health point of view. (P02, academic, law)

However, two participants cautioned that while such a fund was largely welcomed, it would be important that a tobacco control fund did not act as a disincentive for government funding or cutbacks to existing tobacco control activities. Participants also welcomed the fact that an industry-funded payment would help to hold the tobacco industry more accountable for the damage they cause to society, with one participant stating: 'There's some sort of nice symmetry about money from the tobacco industry being used to improve or solve some of the problems it creates' (P05, third sector, public health). However, participants also warned that the funding mechanism of extracting money from the tobacco industry would need careful consideration so that the levy was not passed on to nicotine-dependent and socially deprived smokers, with one participant warning that:

It doesn't really make sense, I think, to pursue further interventions that actually further widen the health inequality that we have. (P07, academic, public health)

### How might a tobacco control fund be designed?

Participants considered in more detail how a tobacco control fund might be designed to raise funds: (1) general excise tax on retail tobacco sales; (2) ring-fenced hypothecation of excise taxes on retail tobacco sales; and (3) a direct levy on tobacco manufacturers. Participants were asked to consider the relative merits of each funding approach.

### General excise tax on retail tobacco sales

Participants were widely supportive of excise taxes, predominantly not only for their role in decreasing consumption but also for their role in fundraising with some participants drawing on excise tax in Australia and New Zealand as useful models for the UK policy-makers to consider. Participants highlighted the simplicity, efficiency and political acceptability of excise tax as positive attributes of this approach. Some participants expressed doubt about the usefulness of excise taxes to fundraising given falling revenue with one academic stating:

The UK I know is now sort of sitting at the top of that revenue situation where they increased excise tax, revenues are not increasing all that much because the excise taxes are very high already. (P01, academic, economics)

In contrast, other participants suggested that the government can effectively control the extent to which taxes are passed on to consumers by capping retail prices, meaning that increasing specific excise tax can raise revenue while ensuring that retail prices do not increase so consumers do not bear the additional cost.[19]

### Ring-fenced hypothecated excise tax

Participants discussed the potential for some or all of excise taxation on retail tobacco sales to be hypothecated, meaning that it would be diverted into a specific fund instead of general government funds. This approach was viewed by participants as publicly acceptable as explained:

Dealing with the consequences or addressing the harms that arise from the product I think is actually instinctively appealing to people. (P22, public sector, public health)

However, while appealing in principle, participants overwhelmingly indicated that hypothecation would meet with too much opposition from HM Treasury, as noted by one academic:

Politicians in general don't like it, they're very particular about being elected to do the right thing, and they wish to retain their independence and their freedom for manoeuvre. So, it can be a challenging negotiation that one. (P03, academic, public health)

Despite this view, participants identified the UK's SDIL as a possible route to hypothecation but noted that the funds raised by SDIL were not ultimately ring fenced for the purposes they originally presented to the public, with one participant noting:

The sugar tax was pushed through with major public support on the basis of hypothecation. And then guess what? There was a crisis and the money, the revenues raised for the sugar tax miraculously didn't get spent on children's breakfast clubs and school sports but have been used to fill gaps in broader public health, and possibly National Health Service (NHS) budgets as well. That's always a risk. (P03, academic, public health)

This led to discussions about how to win political support for hypothecation and the merits of creating a general health fund instead of a tobacco control fund. As explained by one participant:

I absolutely expect that it would be easier to convince policymakers, who generally don't like hypothecated taxes, [of the merits of a general health fund] so the more freedom that you give them, the more acceptable it's likely to be. But I would rather think it probably would be less acceptable to the public, because if you're using the sort of, polluter pays type principle, then, you know, people expect that there is a direct consequence between those two things. (P22, public sector, public health)

Another potential route to hypothecation discussed that bypassed HM Treasury was:

If it was seen as a user fee done by the Department of Health and Social Care, then it would bypass the treasury's normal functioning. (P09, academic, economics)

Bypassing HM Treasury was advocated by three participants who suggested taking inspiration from the UK's Pharmaceutical Price Regulation Scheme (PPRS),[20 21] through which the Department of Health and Social Care (DHSC) (and not the treasury) receives excess profits from participating pharmaceutical companies and uses that funding to address shortfalls in NHS budgets due to expenditure on novel treatments. While this scheme is not an example of hypothecated tax, it was presented as a precedent for the DHSC receiving funds from industry, and an illustration of relevant administrative expertise within the DHSC, as explained:

> The important thing from the tobacco point of view is, you've established this principle of soft hypothecation where the rebates from the industry go back specifically for or back to the [DHSC], rather just the Treasury who just grab it. (P24, private sector, pharmaceuticals)

### A levy on the tobacco industry

Participants who favoured this approach (n=22, 92%) typically viewed it as a means to extract funds from industry instead of from consumers, which may be more appealing to the public and could help convince policy-makers:

> I think politically it's more sellable to the public [than excise tax increases]. (P08, third sector, other)

> I think that would be a decision-making factor for any governmental policy measure that got put forward, that it would be very much clear that the industry would be the contributor, not the public, if you like. (P22, public sector, public health)

Controlling retail prices was deemed an essential part of ensuring that the cost of a levy is borne by the industry.

> The way that the tobacco companies are monopolies and making excess profits is because they are using gradual escalator duty increases to increase their own prices. So, you need to cap prices (P24, private sector, pharmaceuticals).

Conversely, some participants argued that retail prices should not be limited as price increases are beneficial in reducing consumption: as explained:

> When you do see tax increases, you tend to see over-shifting of the tax and using that as an opportunity to raise price and capitalise on at least the addicted consumers that are still in the market. So that is happening, but I don't know that that's necessarily a bad thing, because in the end those price increases are also very effective and leading to additional cessation and particularly in terms of preventing initiation. (P04, academic, economics)

The PPRS was presented as a potential model for extracting industry profits outside of excise taxes, and one that has been refined over many years to limit potential loopholes. However, PPRS was generally considered to be of limited use in having real-world transferability from pharmaceuticals to tobacco. One public sector participant stated:

> The UK pharmaceutical market's status as a virtual monopsony differs starkly from the tobacco industry and suggested that such a scheme may incentivise the lowering of tobacco prices. (P14, public sector, other)

In thinking about general principles of where the tobacco control fund might come from, participants discussed considered three options from 'profits', 'sales volume' or 'market share'. The option of a payment coming directly from 'profits' was largely discounted on the grounds that:

> Multinational companies are very good at moving money around and shifting profits to other countries with lower tax systems. (P03, academic, public health)

The option of using 'sales volume' or 'market share' were both more popular as they were deemed more difficult for companies to obscure and shift money. Examples given were: 'The harm is linked to the sales volume of the product, not to the profits they make from it' (P23, third sector, other). Or that: 'Market share is the easiest way to do it. And you may want to average market share over the past 30 years or something like that to try to figure out what it is' (P02, academic, law).

### Other policy design considerations

After considering the different options for designing a tobacco control fund, participants considered other factors that would be essential for gaining public and political support. A key factor identified was the need for the fund to be administered by a government body with an independent advisory group to ensure transparent decision-making. As highlighted by one participant, a requirement would be:

> A transparent body that both industry and [academic] researchers and the government had trust in to operate transparently and fairly and not be unduly influenced by any stakeholders, you just need to make it an independent body. (P05, third sector, economist)

It was also agreed that the fund should be allocated yearly with distribution at local, regional and national levels to support smoking prevention and cessation rather than treatment activities with priority given to tackling smoking-related inequalities in the most deprived areas. This was deemed important for:

> Making smoking obsolete, to massively benefit the most deprived communities both economically as well as in health terms. (P18, public sector, public health)

## DISCUSSION

Experts considered three broad approaches to raising funds: raising existing excise tax on tobacco sales, introducing a hypothecated excise tax and a tobacco industry levy. Each approach was assessed as having strengths and weaknesses, for example, raising excise taxes was seen as politically feasible and administratively simple, while hypothecation was seen as least politically plausible due to potential Treasury resistance and a tobacco levy was deemed as a logical advocacy route following the polluter pays principle to ensure the industry pays for its damage to society. Experts agreed that whichever mechanism is chosen, must be clearly guided by what the fund is directly trying to achieve. This is consistent with a recent Public Health England report on fiscal and pricing policies,[22] which highlights that policy success depends on the clarity of policy goals. Most experts agreed that key principles underlying the design of a fund would be to collect predictable and reliable funds from transnational tobacco producers either from companies' sales volume or their market share as a way to assign responsibility and establish the proportion they should pay. There was agreement that any fundraising mechanism which extracts funds from industry and avoids the potentially regressive effects of price increases on consumers may be the optimal fundraising approach. However, there was acknowledgement that policy goals have trade-offs. For example, to achieve both health promotion and revenue-raising objectives is possible within the same policy when demand for a product is relatively price inelastic, as is the case with tobacco.[22] From this perspective, permitting costs to be passed on to customers and ensuring that costs are paid by industry may each be valid goals, and designing the policy requires skill. The implementation of other fiscal interventions to tackle the harms caused by unhealthy products, such as the SDIL, our research has shown the political willingness to establish a tobacco control fund. Experts described the potential for resistance from the Treasury, politicians and the public to these three potential tobacco control fund proposals. Industry resistance and influence is relevant in terms of both policy acceptability and ensuring compliance with WHO Framework Convention Tobacco Control,[23] thus future research could explore the potential for resistance from industry actors concerning a design of a tobacco control fund in the UK.

In considering the policy approaches to raise funds lessons may be learnt from other countries such as Australia, Thailand, Vietnam and Korea who have implemented this system for health promotion.[10 11 24] Australia have been leaders in establishing and administering tobacco control funding. The Victorian Health Promotion Foundation (VicHealth), established in 1987, was the first foundation to be funded by a tax on tobacco with a legislative mandate to promote health in the state of Victoria, Australia.[25 26] The levy was set at 5% and this increased the state tobacco licence fee from 25% to 30%.[27] In the first year, the money raised approximately AUS$23 million and this was paid directly into the foundation.[27] It was regarded an inspiration[25 28] and subsequently, led to the establishment of the West Australian Health Promotion Foundation (Healthway).[25] Relevant here is that VicHealth is a self-governing statutory board enabling it to be an independent board. This independence allows the foundation to distance themselves from tobacco industry influence.

Similar to VicHealth, the Thai Health Promotion Foundation (ThaiHealth) is a self-governing statutory board funded by industry money but independent from tobacco industry interference.[25] Revenue for ThaiHealth was established from a new 2% earmarked tax on tobacco and alcohol importers and manufacturers to support tobacco control and health promotion efforts.[29] Vietnam and Korea have also adopted similar funding models[30] mobilising financial resources to strengthen cessation services and develop interventions to help tobacco growers change their occupations.[25 31 32]

In the UK context, there was good agreement that the fund should be focused on tackling smoking-related health inequalities and preventing people from starting to smoke and helping them to quit rather than treating smoking-related diseases. Experts in this current study also suggested that the fund should be ring fenced and allocated yearly with distribution at local, regional and national levels to support a comprehensive tobacco control plan towards meeting government targets. This is similar to the way VicHealth operate where their goals are aligned with government targets for example the 10-year goal that 400 000 more Victorians would be tobacco free by 2023.[33] Experts in this study also identified that the fund should be run in an independent and transparent way without any interference or input from the tobacco industry as VicHealth and ThaiHealth have done.

Several limitations in this study are worth noting. The qualitative nature of data offers depth of opinion within the research sample but does not offer any predictions about the frequency of specific stances within any wider population. We were satisfied that the diversity of professional experience and expertise across these 24 participants provided us with a sample that represented the breadth of perspectives likely to be found within our target population. The value of qualitative policy research is in identifying useful reasoning and novel ideas, not making generalisations about how commonplace specific opinions are. This study was also affected by certain limitations inherent to policy research. The complexity of policies and policy-making environments is such that transferring learning from one policy to a different policy is challenging.[34] For example, the US tobacco Master Settlement Agreement may contain valuable lessons for tobacco control policy in the UK, but the importance of the differences in time periods and legislative contexts cannot be discounted. As such, few participants possessed the breadth of context and knowledge to be able to present comprehensive recommendations for policy. More commonly, participants presented

in-depth knowledge in specific areas or general principles for policy-making. However, this study offers new insights into an under-researched area. While the interviews were valuable in producing rich individual accounts into relevant aspects of tobacco control, the key benefit of the discussion groups was in creating an informed dialogue between experts. Together, this data offered a valuable means of arriving at grounded policy recommendations through interdisciplinary discussion, useful in policy research due to the extent to which policy is constructed through the discursive engagement of different coalitions of policy actors.[35] Another strength was using online data collection which proved to be straightforward reduced geographical barriers to participation among world-leading experts in the UK, the USA and South Africa.

## CONCLUSION

Smoking remains a leading preventable cause of death and disease in the UK with much of this impacting the poorest communities. The implementation of a tobacco control fund would help meet the proposed government tobacco-free targets. However, there is no 'one-size-fits-all' template for such a fund, the structure and operations of the fund would need to adapt to other countries to fit the culture, government ideology and social context. This research shows that experts support the introduction of a tobacco control fund to reduce inequalities in health and achieve the English and Scottish targets of reducing adult smoking prevalence to 5% by 2030 and 2034, respectively. It provides early insights into how a fund might be established and administered in the UK and sets out key foundational principles that must be engaged with in designing a tobacco control fund policy in the UK. Importantly, although there was no one funding approach had unanimous support, experts agreed that establishing an 'imperfect policy' that provides dedicated funding is preferable to delay and inaction.

**Contributors** SH: Conceptualisation, data curation, methodology, validation, writing—original draft preparation and gurantor. MJS: Data curation, visualisation, writing—review & editing. CHB: Conceptualisation, data curation, methodology, investigation, validation, writing—review & editing. CP: Conceptualisation, data curation, methodology, investigation, validation, writing—review & editing.

**Funding** This work was supported by the Medical Research Council (MC_UU_00022/2 and (MC_UU_12017/15), Chief Scientist Office of the Scottish Government Health Directorates (SPHSU15 and SPHSU17) and Cancer Research UK Grant PPRCTAGPJT\100003.

**Competing interests** None declared.

**Patient and public involvement** Patients and/or the public were not involved in the design, or conduct, or reporting, or dissemination plans of this research.

**Patient consent for publication** Not applicable.

**Ethics approval** This study involves human participants and was approved by the college of social sciences research ethics committee at the University of Glasgow (reference number: 400190213). Participants gave informed consent to participate in the study before taking part.

**Provenance and peer review** Not commissioned; externally peer reviewed.

**Data availability statement** All data relevant to the study are included in the article or uploaded as online supplemental information.

**ORCID iDs**
Shona Hilton http://orcid.org/0000-0003-0633-8152
Marissa J Smith http://orcid.org/0000-0002-5017-6085
Chris Patterson http://orcid.org/0000-0002-5914-8718

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
