## [Reviewer comments · BMJ Open]

ARTICLE DETAILS

TITLE (PROVISIONAL)	Experts' views on how to design a Tobacco Control Fund in the UK
AUTHORS	Hilton, Shona; Smith, Marissa; Buckton, Christina; Patterson, Chris

VERSION 1 – REVIEW

REVIEWER	Branston, J. Robert University of Bath, School of Management
REVIEW RETURNED	02-Sep-2022

GENERAL COMMENTS	Thank you for the opportunity to read this interesting paper. Having read the paper through (and drafted my review) I noticed there was an author response document and tracked changes version after the clean version I read. I did not access these as I did not want them to influence my review but it does mean that I want to say that I am sorry for the authors that they are now getting another review from me as an entirely new reviewer, having already had to respond to various reviewer comments at an earlier stage of the process. However, I do like the paper and think it can make a good contribution to the literature once a few further revisions are made. All of my comments are small in nature, and I offer two general comments before a list of more specific points. The title of the article doesn't entirely align with the objective of it as described in the objective section of the abstract, or indeed, as articulated in the introduction. The paper is essentially about the idea of creating a tobacco control fund, but that isn't the same thing as a smokefree generation (or even being smokefree). Given this, I suggest the title be changed so that it better reflects the control fund focus. My only other over-arching comment is that I think the paper could better include citations to more of the tobacco control literature. There were several ideas/concepts mentioned which were not fully explained and which readers might not be familiar with (e.g. Pharmaceutical pricing scheme or how tobacco price capping would work). Including references for such concepts would help readers access the required details. I appreciate the paper isn't about explaining the ideas per se so a full literature review isn't appropriate, but I think offering a bit more integration in the literature at various points would strengthen the paper and readers perceptions of that, by showing it is embedded in the area being discussed. I have below given two particular examples, but I am sure there are other places where an additional reference or two would be helpful too.
---

	Good luck with the revisions! Specific Points P4, line 7. The 2030 target is for England only, not the UK as a whole. Health is a devolved matter in the UK, which means the Westminster government only deals with English health matters. P4, line 23. See earlier comment about the smokefree being an English not UK target. P5, line 17 – see earlier comment on UK vs England. P5, line 21 “which may require additional policy to support tobacco control measures” – unless I am not getting a subtle point, wouldn’t “which may require additional tobacco control measures” be more straightforward? P5, line 24 – Is a tobacco control fund really under consideration? It has certainly been proposed by various groups, but I’m not sure how far that has gotten in official policy circles and hence if it is ‘under consideration’. P6, table 1 – I feel slightly uncomfortable with the way that you have set up your table as you don’t allow for individuals to sit in multiple columns. That I think is important as, for example, an economist who works on tobacco control, might have a different point of view to an economist who works on other sectors generally. I appreciate you want to have a table that sums up to the 24 interviews that you did though. If you want to maintain this approach, I think you could at least identify who you classified individuals – was it their self-selection or was it your categorisation? If the latter, on what basis did you categorise? P.7, line 34. In the analysis section you mentioned the data was placed into inductive categories. Are these categories then used to present the results section? If so I think that should be said explicitly. If not, I think you should explain where the structure of the results section comes from. P.8, line 51. The section on “How might a tobacco control fund be designed?” has a slightly odd order. The second paragraph talks about options for funding, but this comes after the first paragraph that seems to offer some conclusion that market share was the way to do it. I suggest the first and 2nd paragraph be switched. P.9, line 56. I find the idea of capping retail prices is interesting (and have written about it!), but it is not widely considered in tobacco control. I would suggest it could be usefully referenced to show the academic rationale behind the suggestion for any reader interested. I say this not as a shameless attempt to get a citation to my work but in relation to my wider point above about linking several concepts back to the tobacco control literature. P.10, bottom/top of P.11 – the text at the start of the section “A levy on the tobacco industry” repeats itself.
--	--

	P.11, line 20 – I think it would be helpful to offer some references that explain the PPRS – many readers won't have heard of that before and may want further details on how it works. P.12, lines 39-45. I appreciate this is a quote but it is very wrong to say that tobacco is already a "pretty unprofitable product" – actually the manufacturers make massive profits with high margins. Only retailers make small profits selling tobacco so I think some qualification after the quote would be appropriate. P.13, line 56. I believe a full stop is missing after 'levy'. P.14, line 42. "...administering a tobacco control funding". – I believe it should be either '...a control fund', or 'administering tobacco control funding'. P.14, lines 42-50. Can you give some detail on the tax that is applied in the Australian example you cite? I didn't think there was currently any state level taxation in Australia
--	---

REVIEWER	Mamudu, Hadii East Tennessee State University, Health Services Management & Policy
REVIEW RETURNED	06-Oct-2022

GENERAL COMMENTS	TOPIC: How to fund a generation free from tobacco in the UK? Expert views on designing a Tobacco Control Fund. GENERAL COMMENTS: Using qualitative data from expert interviews (N=24) and focus group discussions (N=2), the authors explore views on potential value of, and approach to establishing and administering tobacco control fund. Although tobacco use imposes adverse health and economic costs to individuals and communities, the tobacco industry continues to be very profitable. In contrast, tobacco control aims to alleviate or mitigate the burdens imposed by tobacco use. As such, exploring how to fund tobacco control is of public health significance. My comments are below:  1. Please spell out abbreviations and acronyms. In particular, the introduction contains abbreviations that may be only familiar to people residing in UK and Scotland, e.g., "HM Treasury". 2. Can the authors quantify some of the presented information? This is a semi-structured interview; therefore, the authors can indicate the % of people who support levy on the tobacco industry, for example. SPECIFIC COMMENTS: Abstract: Clearly written and well-constructed; however, I will recommend that the authors should present the findings in the qualitative data, instead of saying "The experts suggest ...". Please remove the word "suggests" and present the results of the analysis. Introduction: It is well-written with clear purpose. Methods:
---

	It is standard and appropriate. However, the authors should:  1. Clear indicate the analytic approach – grounded theory, content analysis, phenomenology etc. used in the study 2. Provide more information on the discussion groups --- how many participants were in each group etc. 3. Explain why the interviews were limited to 24 participants? 4. Clarify if all members of the research team were involved in the data management – coding etc. Results: The results are well-presented and readable. However, as a policy analysis paper, this section can be strengthened with some quantification of the results (see comment above). Discussion: Results were discussed with relevant literature in the region where the study was conducted. Made the case for the suggestions in the conclusion. Conclusion: It is clearly written; however, I am not sure if all the information presented emanates from the results. For e.g., the authors make assertion that “The UK is a leader of tobacco control....” Where did this information come from in the results? Nonetheless, such an assertion can be contested. For example, countries such as New Zealand are pursuing Endgame Strategies, and this idea of tobacco control fund is being borrowed from places such as Australia. Therefore, I will urge the authors to focus the conclusion exclusive on issues that emanate from the results. MINOR COMMENTS:  1. P.14, ln 46: please fix the repetition – “... policy to a different”
--	---

REVIEWER	Wagner-Rizvi, Tracey The University of Edinburgh School of Social and Political Science
REVIEW RETURNED	06-Oct-2022

GENERAL COMMENTS	While you have described the potential for resistance from the Treasury and politicians as well as from the public to these three potential tobacco control fund proposals, you do not mention the potential for resistance from industry actors. This is surprising given your citation (though not in an appropriate context) of Smith, Savel and Gilmore (2013), who examine exactly such tactics. Industry resistance and influence is relevant in terms of both policy acceptability and ensuring compliance with FCTC Article 5.3 (both while designing and establishing a tobacco control fund, and in ensuring the independence of the administrating government body and the advisory group). If you have data relating to this from your interviews and discussion groups, this seems essential to include in the article. If such data is not available, you might address this in the Discussion (Was it not questioned in the interviews and discussion groups? Was it not a concern for the experts? What might that suggest?), and consider whether this can or should be addressed in subsequent research. To ensure the novelty/timeliness of the article does not get lost, it would be good to point again in the discussion and/or conclusion to the potentially increased political willingness to establish a
---

	tobacco control fund following the introduction of other fiscal interventions to tackle harms caused by unhealthy products, especially the SDIL. The article is clearly written but would benefit from careful proof-reading and formatting. For example (not-exhaustive): Page 2, line 9: value of, and approaches to; line 24: academic, public, private and third sectors; line 42: would be a valuable method; lines 44-45: either using either companies' sales volume or market share Page 5, line 11: academic, public, private and third sectors; line 37: and 60 minutes. All were recorded Page 6, line 53: replace identifiable details; line 54: granted on by the College Page 7, line 25: cautioned that whilst such a fund; line 31: payment, would help hold; lines 57-59: participants discussed considered three options: from 'profits', 'sales volume' What does the abbreviation MSA (page 14, line 49) stand for? Review the formatting of quotations, especially when and how to use block text and when to include in-line. Check for consistency in using italics for quotations and their brackets and where punctuation goes around citations. Remove spaces from before the first word of several paragraphs.
--	---

VERSION 1 – AUTHOR RESPONSE

Reviewer 1

R1.1 The title of the article doesn't entirely align with the objective of it as described in the objective section of the abstract, or indeed, as articulated in the introduction. The paper is essentially about the idea of creating a tobacco control fund, but that isn't the same thing as a smokefree generation (or even being smokefree). Given this, I suggest the title be changed so that it better reflects the control fund focus.

- *We thank the reviewer for this comment and agree that the title should be amended to align with the objective of the study. We have revised the title to:*

"Experts' views on how to design a Tobacco Control Fund in the UK"

R1.2 My only other over-arching comment is that I think the paper could better include citations to more of the tobacco control literature. There were several ideas/concepts mentioned which were not fully explained and which readers might not be familiar with (e.g., Pharmaceutical pricing scheme or how tobacco price capping would work). Including references for such concepts would help readers access the required details. I appreciate the paper isn't about explaining the ideas per se so a full literature review isn't appropriate, but I think offering a bit more integration in the literature at various points would strengthen the paper and readers perceptions of that, by showing it is embedded in the area being discussed. I have below given two particular examples, but I am sure there are other places where an additional reference or two would be helpful too.

- *We thank the reviewer for this comment and agree that inclusion of citations when discussing specific concepts (e.g., pharmaceutical pricing scheme) would strengthen the paper. We have revised the manuscript to include citations, please see our tracked changes on the manuscript.*

R1.3 P4, line 7. The 2030 target is for England only, not the UK as a whole. Health is a devolved matter in the UK, which means the Westminster government only deals with English health matters.

- *We thank the reviewer for this comment. We have revised the manuscript (see below) to ensure we are referring to England and Scotland, not the UK.*

“To further reduce smoking, the Westminster and Scottish governments have proposed targets to reduce adult smoking prevalence to 5% by 2030 and 2034, respectively [1, 2].”

R1.4 P4, line 23. See earlier comment about the smokefree being an English not UK target.

- *We thank the reviewer for this comment. We have revised the manuscript to ensure when discussing smokefree that this is an English, not a UK target.*

R1.5 P5, line 17 – see earlier comment on UK vs England.

- *Please see our response to R1.4.*

R1.6 P5, line 21 “which may require additional policy to support tobacco control measures” – unless I am not getting a subtle point, wouldn’t “which may require additional tobacco control measures” be more straightforward?

- *We thank the reviewer for this comment, and we agree with the reviewers suggested change. We have revised the manuscript as follows:*

“However, for these proposed targets to be met prevalence rates need to decline at a much faster rate [3] which may require additional tobacco control measures.”

R1.7 P5, line 24 – Is a tobacco control fund really under consideration? It has certainly been proposed by various groups, but I’m not sure how far that has gotten in official policy circles and hence if it is ‘under consideration’.

- *We thank the reviewer for their comment. At present Cancer Research UK are interested in ‘testing the water’ but we are not privy to knowing whether government has it on the agenda. We have revised the manuscript to remove mention of ‘under consideration’, see our amendments below.*

“One policy option that has been proposed is to establish a ring-fenced tobacco control fund.”

R1.8 P6, table 1 – I feel slightly uncomfortable with the way that you have set up your table as you don’t allow for individuals to sit in multiple columns. That I think is important as, for example, an economist who works on tobacco control, might have a different point of view to an economist who works on other sectors generally. I appreciate you want to have a table that sums up to the 24 interviews that you did though. If you want to maintain this approach, I think you could at least identify who you classified individuals – was it their self-selection or was it your categorisation? If the latter, on what basis did you categorise?

- *We thank the reviewer for their comment. All of the participants were involved in tobacco control research, advocacy or policy. The categorisation based on people’s professional roles. The economists and legal scholars were all people who were (or had) actively been involved in tobacco control through those disciplines. We have renamed the ‘tobacco control’ column to ‘other’, please see the manuscript for our tracked changes. The ‘other’ column covers people from public health organisations who had involvement in tobacco control policy and people from academia who did research and advocacy around tobacco control, but not from a legal or economics position.*

R1.9 P.7, line 34. In the analysis section you mentioned the data was placed into inductive categories. Are these categories then used to present the results section? If so, I think that should be said explicitly. If not, I think you should explain where the structure of the results section comes from.

- *We thank the reviewer for their comment. We can confirm that the inductive categories were used to present the results and we have revised the manuscript to indicate this.*

“The results are presented in accordance with the inductive coding categories developed during the analysis stage.”

R1.10 P.8, line 51. The section on “How might a tobacco control fund be designed?” has a slightly odd order. The second paragraph talks about options for funding, but this comes after the first paragraph that seems to offer some conclusion that market share was the way to do it. I suggest the first and 2nd paragraph be switched.

- *We thank the reviewer for their comment and agree that the ordering of the section is confusing. As suggested, we have moved the first paragraph to after the second paragraph. Please see the manuscript for tracked changes.*

R1.11 P.9, line 56. I find the idea of capping retail prices is interesting (and have written about it!), but it is not widely considered in tobacco control. I would suggest it could be usefully referenced to show the academic rationale behind the suggestion for any reader interested. I say this not as a shameless attempt to get a citation to my work but in relation to my wider point above about linking several concepts back to the tobacco control literature.

- *We thank the reviewer for their comment and agree referencing other literature (see text below) relating to capping retail prices would strengthen the paper and be beneficial to the reader.*

“In contrast, other participants suggested that the government can effectively control the extent to which taxes are passed on to consumers by capping retail prices, meaning that increasing specific excise tax can raise revenue while ensuring that retail prices do not increase so consumers do not bear the additional cost [4].”

R1.12 P.10, bottom/top of P.11 – the text at the start of the section “A levy on the tobacco industry” repeats itself.

- *We thank the reviewer for raising this point. We have revised the manuscript to amend repetition.*

R1.13 P.11, line 20 – I think it would be helpful to offer some references that explain the PPRS – many readers won’t have heard of that before and may want further details on how it works.

- *We thank the reviewer for their comment. We have amended the manuscript to include citations relating to PPRS.*

“Bypassing HM Treasury was advocated by three participants who suggested taking inspiration from the UK’s Pharmaceutical Price Regulation Scheme (PPRS) [5, 6], through which the Department for Health and Social Care (DHSC) (and not the treasury) receives excess profits from participating pharmaceutical companies and uses that funding to address shortfalls in NHS budgets due to expenditure on novel treatments.”

R1.14 P.12, lines 39-45. I appreciate this is a quote but it is very wrong to say that tobacco is already a “pretty unprofitable product” – actually the manufacturers make massive profits with high margins. Only retailers make small profits selling tobacco so I think some qualification after the quote would be appropriate.

- *We thank the reviewer for their comment. We have revised the manuscript by remove this quote.*

R1.15 P.13, line56. I believe a full stop is missing after 'levy'.

- *We have amended the manuscript.*

R1.16 P.14, line 42. "...administering a tobacco control funding". – I believe it should be either '....a control fund', or 'administering tobacco control funding'.

- *We thank the reviewer for this comment and have revised the manuscript as follows.*

"Australia have been leaders in establishing and administering tobacco control funding."

R1.17 P.14, lines 42-50. Can you give some detail on the tax that is applied in the Australian example you cite? I didn't think there was currently any state level taxation in Australia

- *We thank the reviewer for their comment. We have revised the manuscript as follows:*

"The Victorian Health Promotion Foundation (VicHealth), established in 1987, was the first foundation to be funded by a tax on tobacco with a legislative mandate to promote health in the state of Victoria, Australia [7, 8]. The levy was set at 5% and this increased the state tobacco licence fee from 25% to 30%. In the first year, the money raised approximately \$23 million and this was paid directly into the foundation [9]."

Reviewer 2

R2.1 Please spell out abbreviations and acronyms. In particular, the introduction contains abbreviations that may be only familiar to people residing in UK and Scotland, e.g., "HM Treasury".

- *We have revised the manuscript to spell out all abbreviation and acronyms used.*

R2.2 Can the authors quantify some of the presented information? This is a semi-structured interview; therefore, the authors can indicate the % of people who support levy on the tobacco industry, for example.

- *We thank the reviewer for this comment. We have revised the manuscript as follows:*

"Where appropriate, the number of participants that gave specific opinions are presented as counts and proportions to help illustrate the balance of opinion with the sample. However, it must be noted that, given the qualitative methodology used in this study, these numbers cannot necessarily be generalised to any wider population."

"Participants who favoured this approach (n=22, 92%) typically viewed it as a means to extract funds from industry instead of from consumers, which may be more appealing to the public and could help convince policymakers."

R2.3 Abstract: Clearly written and well-constructed; however, I will recommend that the authors should present the findings in the qualitative data, instead of saying "The experts suggest ...". Please remove the word "suggests" and present the results of the analysis.

- *We thank the reviewer for this comment. We have revised the results section of the abstract as follows:*

"Most experts agreed that a ring-fenced tobacco control fund would be a valuable method of raising predictable and reliable funds from tobacco producers either using either companies' sales volume or market share as a way to establish the proportion they should pay.

Experts predominantly recommended that a fund in the UK should be administered by a government body with devolved nation input and with an independent advisory group. They typically indicated that funding should be allocated yearly with a distribution at local, regional, and national

levels to support smoking prevention and cessation rather than treatment activities with priority given to measures that tackle smoking-related inequalities.”

Methods: It is standard and appropriate. However, the authors should:

R2.4 Clear indicate the analytic approach – grounded theory, content analysis, phenomenology etc. used in the study

- *We thank the reviewer for this comment. We have revised the manuscript to indicate the analytical approach.*

“We conducted thematic analysis of the data from the interview transcripts and discussion group minutes. The process followed Braun and Clarke’s [10] six-phase framework for thematic analysis. The research team read and re-read the transcripts to become familiar with the data, and then iteratively constructed a coding frame to enable consistent organisation of relevant data. NVivo was used to organise categories on the basis of inductive themes that emerged from close reading of the, capture of both areas of agreement and less typical perspectives across a range of categories. The discussion group recordings and minutes were cross-compared with the interview coding frame to confirm and expand on codes relating to recommendations for policy design of a tobacco control fund.”

R2.5 Provide more information on the discussion groups --- how many participants were in each group etc.

- *We thank the reviewer for this comment and we have revised the manuscript (see below) to indicate the number of participants in each discussion group.*

“In March 2021, two follow-up online discussion groups were conducted with nine individuals using Microsoft Teams. Participants were selected for these follow-ups based on their sectorial expertise and to represent key disciplines. The first discussion group (n=5) included three third sector professionals with expertise spanning tobacco control and public health advocacy and two academic economists. The second discussion group (n=4) included two public sector professionals with roles in tobacco control and public health policy, and two academics with expertise in law and public health.”

R2.6 Explain why the interviews were limited to 24 participants?

- *We thank the reviewer for this comment. The project had limited funding, thus we had to recruit within the scope of the budget. We have revised the manuscript as follows:*

“We were satisfied that the diversity of professional experience and expertise across these 24 participants provided us with a sample that represented the breadth of perspectives likely to be found within our target population.”

R2.7 Clarify if all members of the research team were involved in the data management – coding etc.

- *We thank the reviewer for their comment. We have revised the manuscript (see tracked changes) to indicate what members for the research team were involved in each task. We have also revised the contributions section at the end of the manuscript (see below) to follow the CRediT classification.*

“**Contributions: Shona Hilton:** Conceptualisation, Data Curation, Methodology, Validation, Writing-Original draft preparation. **Marissa J. Smith:** Data Curation, Visualisation, Writing - Review & Editing. **Christina Buckton:** Conceptualisation, Data Curation, Methodology, Investigation, Validation, Writing - Review & Editing. **Chris Patterson:** Conceptualisation, Data Curation, Methodology, Investigation, Validation, Writing - Review & Editing.”

Results:

R2.8 The results are well-presented and readable. However, as a policy analysis paper, this section can be strengthened with some quantification of the results (see comment above).

- *In response to R2.2 and R2.5 we have include some quantifying details. Please see our responses to R2.2 and R2.5 for included details and the manuscript for tracked changes.*

Conclusion:

R2.9 It is clearly written; however, I am not sure if all the information presented emanates from the results. For e.g., the authors make assertion that “The UK is a leader of tobacco control....” Where did this information come from in the results? Nonetheless, such an assertion can be contested. For example, countries such as New Zealand are pursuing Endgame Strategies, and this idea of tobacco control fund is being borrowed from places such as Australia. Therefore, I will urge the authors to focus the conclusion exclusive on issues that emanate from the results.

- *We thank the reviewer for their comment. We have revised the manuscript to focus on the results from this research.*

“The implementation of a tobacco control fund would help meet the proposed government tobacco-free targets.”

R2.10 P.14, ln 46: please fix the repetition – “... policy to a different”

- *We have revised the manuscript to amend repetition.*

Reviewer 3

R3.1 While you have described the potential for resistance from the Treasury and politicians as well as from the public to these three potential tobacco control fund proposals, you do not mention the potential for resistance from industry actors. This is surprising given your citation (though not in an appropriate context) of Smith, Savel and Gilmore (2013), who examine exactly such tactics. Industry resistance and influence is relevant in terms of both policy acceptability and ensuring compliance with FCTC Article 5.3 (both while designing and establishing a tobacco control fund, and in ensuring the independence of the administrating government body and the advisory group). If you have data relating to this from your interviews and discussion groups, this seems essential to include in the article. If such data is not available, you might address this in the Discussion (Was it not questioned in the interviews and discussion groups? Was it not a concern for the experts? What might that suggest?), and consider whether this can or should be addressed in subsequent research.

- *We thank the reviewer for their comment. Firstly, we have removed Smith, Savel and Gilmore (2013) citation. Secondly, this theme/topic of industry resistance was not raised by participants, possibly because when it comes to tobacco (as opposed to other unhealthy commodities) it is such a broadly accepted (and formalised) norm that the industry cannot be trusted and must not have any involvement in policy. We agree that this could be addressed in future research and have revised the discussion section of the manuscript to include this.*

“Experts described the potential for resistance from the Treasury, politicians and the public to these three potential tobacco control fund proposals. Industry resistance and influence is relevant in terms of both policy acceptability and ensuring compliance with World Health Organisation (WHO) Framework Convention Tobacco Control (FCTC) [11], thus future research could explore the potential for resistance from industry actors concerning a the design of a tobacco control fund in the UK.”

R3.2 To ensure the novelty/timeliness of the article does not get lost, it would be good to point again in the discussion and/or conclusion to the potentially increased political willingness to establish a tobacco control fund following the introduction of other fiscal interventions to tackle harms caused by unhealthy products, especially the SDIL.

- *We thank the reviewer for their comment and we agree that amending the discussion to include mention of political willingness would strengthen the paper. Please see the text below.*

“The implementation of other fiscal interventions to tackle the harms caused by unhealthy products, such as the SDIL, our research has shown the political willingness to establish a tobacco control fund.”

R3.3 The article is clearly written but would benefit from careful proof-reading and formatting. For example (not-exhaustive):

Page 2, line 9: value of, and approaches to; line 24: academic, public, private and third sectors; line 42: would be a valuable method; lines 44-45: either using either companies’ sales volume or market share Page 5, line 11: academic, public, private and third sectors; line 37: and 60 minutes. All were recorded Page 6, line 53: replace identifiable details; line 54: granted on by the College Page 7, line 25: cautioned that whilst such a fund; line 31: payment, would help hold; lines 57-59: participants discussed considered three options: from ‘profits’, ‘sales volume’

- *We thank the reviewer for their comment. We have revised the manuscript to amend the above spelling/grammatical errors and have proofread the full manuscript.*

R3.4 What does the abbreviation MSA (page 14, line 49) stand for?

- *We thank the reviewer for raising this point and we apologise for any confusion. We have revised the manuscript as follows.*

“For example, the US tobacco Master Settlement Agreement (MSA) may contain valuable lessons for tobacco control policy in the UK, but the importance of the differences in time periods and legislative contexts cannot be discounted.”

R3.5 Review the formatting of quotations, especially when and how to use block text and when to include in-line.

- *We have ensured consistency in formatting on quotations.*

R3.6 Check for consistency in using italics for quotations and their brackets and where punctuation goes around citations.

- *We have ensured consistency in italics and punctuation for citations.*

R3.7 Remove spaces from before the first word of several paragraphs.

- *We have revised the manuscript to amend double space before first word.*

References

1. HM Government. Advancing our health: prevention in the 2020s – consultation document 2019 [Available from: <https://www.gov.uk/government/consultations/advancing-our-health-prevention-in-the-2020s/advancing-our-health-prevention-in-the-2020s-consultation-document>].
2. The Scottish Government. Creating a Tobacco-Free Generation: A Tobacco Control Strategy for Scotland. Edinburgh: The Scottish Government; 2013.
3. Cancer Intelligence Team. Smoking prevalence projections for England, Scotland, Wales and Northern Ireland, based on data to 2018/19: Cancer Research UK; 2020 [
4. Branston JR, Gilmore AB. The case for Ofsmoke: the potential for price cap regulation of tobacco to raise £500 million per year in the UK. *Tob Control*. 2014;23(1):45-50.
5. Collier J. The pharmaceutical price regulation scheme. *BMJ*. 2007;334(7591):435-6.
6. Rodwin MA. How the United Kingdom Controls Pharmaceutical Prices and Spending: Learning From Its Experience. 2021;51(2):229-37.
7. International Union Against Tuberculosis and Lung Disease. Sustainable Funding Models for Tobacco Control: a Discussion Paper. 2014.

8. Bonito S. VicHealth: The World's First Health Promotion Foundation. Michigan: University of Michigan; 2014.
9. Victorian Health Promotion Foundation. The Story of VicHealth. Carlton, Australia; 2005.
10. Braun V, Clarke V. Thematic analysis. APA handbook of research methods in psychology, Vol 2: Research designs: Quantitative, qualitative, neuropsychological, and biological. APA handbooks in psychology®. Washington, D.C., USA: American Psychological Association; 2012. p. 57-71.
11. World Health Organisation. The WHO framework convention on tobacco control: an overview. Geneva, Switzerland: World Health Organisation; 2015.